# 🔮 CRYSTAL: Introspective Reasoners Reinforced with Self-Feedback

**Jiacheng Liu**♡♣∗  **Ramakanth Pasunuru**♣
**Hannaneh Hajishirzi**♡♠  **Yejin Choi**♡♠  **Asli Celikyilmaz**♣

♡Paul G. Allen School of Computer Science & Engineering, University of Washington
♣FAIR, Meta   ♠Allen Institute for Artificial Intelligence
liujc@cs.washington.edu

## Abstract

Extensive work has shown that the performance and interpretability of commonsense reasoning can be improved via knowledge-augmented reasoning methods, where the knowledge that underpins the reasoning process is explicitly verbalized and utilized. However, existing implementations, including "chain-of-thought" and its variants, fall short in capturing the *introspective* nature of knowledge required in commonsense reasoning, and in accounting for the mutual adaptation between the generation and utilization of knowledge. We propose a novel method to develop an introspective commonsense reasoner, CRYSTAL. To tackle commonsense problems, it first introspects for knowledge statements related to the given question, and subsequently makes an informed prediction that is grounded in the previously introspected knowledge. The knowledge introspection and knowledge-grounded reasoning modes of the model are tuned via reinforcement learning to mutually adapt, where the reward derives from the feedback given by the model itself. Experiments show that CRYSTAL significantly outperforms both the standard supervised finetuning and chain-of-thought distilled methods, and enhances the transparency of the commonsense reasoning process. Our work ultimately validates the feasibility and potential of reinforcing a neural model with self-feedback. [1]

## 1 Introduction

Commonsense reasoning poses unique challenges to neural reasoning models. The underlying knowledge that grounds the reasoning process is often obscure and inexplicable, even to humans as we mainly rely on intuitive inference for such problems (Mercier and Sperber, 2017). This is in stark contrast with multi-step logical reasoning (e.g.,

---
∗Work done as a visiting researcher at FAIR, Meta.
[1]Code: github.com/liujch1998/crystal
Model: huggingface.co/liujch1998/crystal-11b
Demo: huggingface.co/spaces/liujch1998/crystal

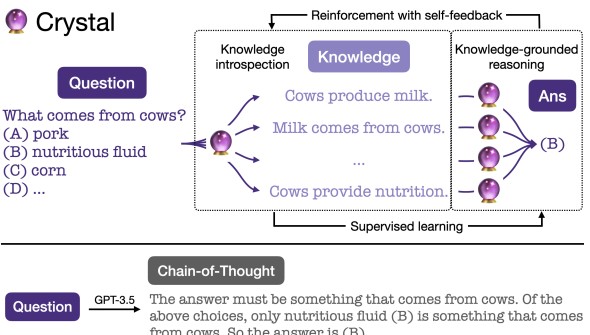

Figure 1: **Top:** CRYSTAL performing introspective reasoning on a commonsense question. The model first uses its knowledge introspection mode to generate relevant knowledge statements, then invokes a knowledge-grounded reasoning mode to predict an answer based on the introspected knowledge. **Bottom:** chain-of-thought prompting on the same question (generated by text-davinci-003 with original few-shot prompts in Wei et al. (2022)). The intermediate steps fail to provide meaningful insight into the reasoning process.

math problems, logical deductions), where the reasoning process consists of explicit and closed-world deduction steps. Chain-of-thought (CoT) (Wei et al., 2022) and its variants have been successful in multi-step logical reasoning, yet their effectiveness on commonsense reasoning is marginal, largely due to the lack of the above observation when designing their few-shot prompts. Nevertheless, generating the reasoning process is still instrumental for commonsense reasoning, as it improves both performance and interpretability of neural models (Shwartz et al., 2020; Liu et al., 2022a, *i.a.*). For such knowledge-augmented reasoning approach, two components are indispensable: (1) *introspecting* for relevant, high-quality knowledge, and (2) effectively and faithfully utilizing the knowledge to make informed final predictions.

Our key insight is that these two components are deeply adaptive to each other: knowledge introspection should aim to produce knowledge that

would be most beneficial to grounding the subsequent reasoning, and knowledge-grounded reasoning should learn to best leverage the previously introspected knowledge. Existing literature does not comprehensively optimize these two components and the bi-directional interaction between them, and comes with additional complications. Knowledge-augmented reasoning methods largely employ task-specific engineering for knowledge generation and are thus difficult to generalize to unseen domains. As for CoT and its variants, the reasoning chains hardly provide meaningful information due to deficiency in their prompt design (Figure 1).

We aim to systematically address the above considerations and build a strong, interpretable and generalizable model for commonsense reasoning. The *introspective reasoner* we develop, named CRYSTAL, tackles commonsense problems by the following (illustrated in Figure 1): it first invokes a *knowledge introspection* mode to generate knowledge statements related to the given question, and subsequently invokes a *knowledge-grounded reasoning* mode that ingests both the question and the previously introspected knowledge to predict an answer. CRYSTAL is trained with reinforcement learning (RL) to improve the synergy between the reasoning paths and the final predictions. The knowledge introspection mode of the model is trained with PPO (Schulman et al., 2017) to optimize a reward function that characterizes if the generated knowledge can fix prediction errors made by the knowledge-grounded reasoning mode of the model. In this sense, CRYSTAL is reinforced with self-generated feedback. Concurrently, the knowledge-grounded reasoning mode evolves to better utilize the introspected knowledge statements for more accurate predictions. These two learning objectives are harmonized through a novel interleaved optimization schedule, echoing the principles of the EM algorithm (Dempster et al., 1977). We employ a two-stage training process: the RL training stage is preceded by a supervised training stage, where CRYSTAL acquires preliminary skills to generate and utilize knowledge by imitating a larger LM (e.g., GPT-3).

Experimental results on 25 commonsense QA benchmarks (10 seen, 15 unseen) show that CRYSTAL not only enhances performance within fixed model sizes, but also amplifies the interpretability of the reasoning process. CRYSTAL outperforms direct QA models finetuned with standard supervised learning and the same data, improving absolute accuracy by 1.5% to 2.5% on different model sizes, and showcases good generalization to unseen benchmarks. This highlights the benefits of introspective reasoning over direct inference. Additionally, CRYSTAL substantially outperforms models distilled from CoT produced by large LMs. Through CRYSTAL, we illustrate the potential and viability of reinforcing neural reasoning models with self-feedback. An additional benefit of our approach is the memory and time-efficient implementation of PPO via model sharing, which allows this state-of-the-art RL algorithm to be applied to larger models with given amount of resources.

## 2 Method

We will first introduce the concept of introspective reasoning (§2.1), followed by a description of our introspective reasoner, CRYSTAL, including its basic functionality (§2.2), training objectives (§2.3), adaptation of the RL algorithm and efficiency improvements (§2.4), and the design of model training process (§2.5, §2.6).

### 2.1 Introspective Reasoning

Conventionally, commonsense QA models are designed to directly predict answers for questions (e.g. Lourie et al., 2021). These models operate like black boxes and their predictions are difficult to interpret. We consider **introspective reasoning**, where a system first introspects for commonsense knowledge statements that are relevant to reasoning about the given question, and subsequently makes an informed prediction that is grounded in the introspected knowledge. We refer to the former step as *knowledge introspection*, and the latter step as *knowledge-grounded reasoning*.

Figure 1 exemplifies the introspective reasoning process. Given the question *"What comes from cows?"* with the correct answer *"nutritious fluids"* provided among other incorrect choices, the system first generates knowledge statements like *"Cows produce milk."* Taking this knowledge statement as additional input, the system then invokes knowledge-grounded reasoning and makes a correct prediction.

Introspective reasoning has promise in enhancing model performance on commonsense reasoning while making the reasoning process more interpretable. The introspected knowledge reveals the

| Mode | I/O Format |
|------|-----------|
| Knowledge introspection | **Input:** `What comes from cows? \n (A) pork (B) can be organic ... (G) nutritious fluid (H) corn \n Knowledge:` 
 **Output:** `Cows produce milk.` |
| Knowledge-grounded reasoning | **Input:** `What comes from cows? \n (A) pork (B) can be organic ... (G) nutritious fluid (H) corn \n Knowledge: Cows produce milk. \n Answer:` 
 **Output:** `G` |

Table 1: CRYSTAL's I/O format for its two modes.

reasoning paths that lead to the final predictions, which human users can observe. The system can explore and ensemble multiple reasoning paths and thus make a more informed final prediction.

**Knowledge introspection.** The term "knowledge introspection" was introduced by Liu et al. (2022a), which also developed a dedicated knowledge introspection model, Rainier. Our work extends this idea to a unified introspective reasoning model.

## 2.2 CRYSTAL

CRYSTAL, the introspective reasoner that we develop, is a unified model that supports the end-to-end workflow of an introspective reasoning system. CRYSTAL has two modes of operation: *knowledge introspection* and *knowledge-grounded reasoning*. In knowledge introspection, the model accepts a question as input, and outputs a knowledge statement relevant to the question. In knowledge-grounded reasoning, the model ingests both the question and the previously generated knowledge statement as input, and outputs a prediction. The system produces an final prediction by consulting and aggregating predictions resulted from all the available reasoning paths, effectively harnessing the power of ensembled reasoning.

**I/O format.** Since CRYSTAL has two modes of operation, it needs to discern when to conduct knowledge introspection and when to engage in knowledge-grounded reasoning. Drawing inspiration from Tafjord and Clark (2021), we structure the input/output format as demonstrated in Table 1. This format is adapted from UnifiedQA (Khashabi et al., 2020), and we detail our modifications in §A.

**Notation.** CRYSTAL is a sequence-to-sequence generative model parameterized by $\theta$. In knowledge introspection, the modeling of knowledge $k$ given question $q$ is denoted as $p_{\text{QK}}(k|q;\theta)$; in knowledge-grounded reasoning, the modeling of answer prediction $a$ is denoted as $p_{\text{QKA}}(a|q,k;\theta)$.

## 2.3 Training Objectives

To yield the desired outcome of an introspective reasoning system, we need to make knowledge introspection and knowledge-grounded reasoning well-adapted to each other. The knowledge introspection component should aim to produce knowledge that would be most beneficial to ground the subsequent reasoning, and the knowledge-grounded reasoning component should learn to best leverage the previously introspected knowledge. We design the training objectives to account for this mutual adaptation.

**Adapting reasoning to introspection.** Suppose a knowledge statement is sampled online from CRYSTAL in the knowledge introspection mode: $\hat{k} \sim p_{\text{QK}}(k|q;\theta)$. We use standard supervision and minimize a knowledge-grounded reasoning loss:

$$\mathcal{L}_{\text{QKA}}(\theta) = -\log p_{\text{QKA}}(a^*|q,\hat{k};\theta),$$

where $a^*$ is the correct answer for question $q$.

**Adapting introspection to reasoning.** The desirability of introspected knowledge is determined by its effectiveness on the subsequent knowledge-grounded reasoning. A knowledge statement is good if grounding in it can remediate an otherwise incorrect prediction, and is bad if it misleads an otherwise correct prediction. Formally, a good knowledge statement should yield

$$a^* \neq \arg\max_{a \in A} p_{\text{QKA}}(a|q,\varepsilon;\theta),$$
$$a^* = \arg\max_{a \in A} p_{\text{QKA}}(a|q,\hat{k};\theta),$$

where $A$ is the candidate set for question $q$, and $\varepsilon$ stands for no knowledge; and vice versa for a bad knowledge statement.

However, a knowledge statement consists of a sequence of discrete tokens, rendering standard gradient methods infeasible for optimizing the introspected knowledge. Following Liu et al. (2022a), we formulate the problem as reinforcement learning (RL), and optimize a reward function that characterizes the desirability of knowledge:

$$r = \frac{1}{2}\Big[\tanh\big(s(a^*|q,\hat{k}) - \max_{a' \in A \setminus \{a^*\}} s(a'|q,\hat{k})\big) \\ - \tanh\big(s(a^*|q,\varepsilon) - \max_{a' \in A \setminus \{a^*\}} s(a'|q,\varepsilon)\big)\Big],$$

where $s(a|q, k)$ is the pre-softmax logit of $p_{QKA}(a|q, k; \theta)$ on the single-token answer $a$. The reward approaches $+1$ for good knowledge statements and $-1$ for bad ones.

We use the PPO algorithm to optimize this reward. A knowledge introspection loss $\mathcal{L}_{PPO}(\theta)$ can be defined as a function of the reward and the model parameter $\theta$, following Liu et al. (2022a). Since this training loss is derived from the downstream knowledge-grounded reasoning results produced by the same model, the model is **reinforced with feedback given by itself**.

During training, the two objectives, $\mathcal{L}_{PPO}(\theta)$ and $\mathcal{L}_{QKA}(\theta)$, are optimized under an interleaved schedule (rather than jointly), which is described in §2.6.

**Leaving out the direct QA objective.** To prevent the model from taking reasoning shortcuts and encourage it to leverage the introspected knowledge, we deliberately left out a potential, direct QA objective:

$$\mathcal{L}_{QA} = -\log p_{QA}(a^*|q). \quad (1)$$

As we will show in experiments, including this direct QA loss hurts performance, probably by allowing the model to take shortcuts around the knowledge.

## 2.4 PPO and Model Sharing

PPO, or Proximal Policy Optimization (Schulman et al., 2017), is an RL algorithm that has been widely used in aligning LMs with human feedback (Stiennon et al., 2020; Ouyang et al., 2022; OpenAI, 2022; Wu et al., 2023). It is also adopted by Liu et al. (2022a) to train their knowledge introspection model, Rainier.

Within the context of PPO terminology, CRYS­TAL's knowledge introspection mode assumes the role of the *policy model*, while its knowledge-grounded reasoning mode functions as the *reward model*. PPO further employs a *value model* to estimate the value function for states containing partial knowledge statements, and we propose to reuse the parameters of CRYSTAL for the value model as well. Consequently, while in conventional PPO the policy, value and reward models are parameterized separately, when training CRYSTAL they share the same underlying parameters. CRYSTAL is essentially a generative LM equipped with two heads: an LM head that comes into play during the policy and reward modeling, and a value regression head that is activated in value modeling. This model sharing

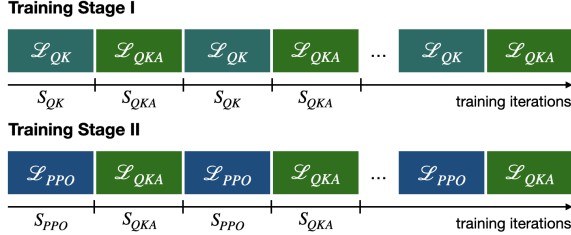

Figure 2: Illustration of the interleaved optimization schedule for both training stages. In training stage I, during each cycle, $\mathcal{L}_{QK}$ is optimized for $S_{QK}$ iterations, and then $\mathcal{L}_{QKA}$ is optimized for $S_{QKA}$ iterations. Progressing to training stage II, during each cycle, $\mathcal{L}_{PPO}$ is optimized for $S_{PPO}$ iterations, and then $\mathcal{L}_{QKA}$ is optimized for $S_{QKA}$ iterations.

results in improved memory and time efficiency for PPO training, as discussed in §4.4.

## 2.5 Two-Staged Training

PPO requires that the policy model is initialized from a reasonably good state. Typically, PPO training follows a supervised finetuning stage for the policy model (Stiennon et al., 2020; Ouyang et al., 2022). For Rainier, an imitation learning stage, during which the model is supervised on *silver* knowledge statements obtained from a few-shot GPT-3, precedes the RL training stage. This imitation learning stage imparts the model with the preliminary skill of generating question-specific knowledge, and sets a promising starting point for RL.

Drawing from this concept, we employ a two-stage training process for CRYSTAL. In training stage I, the model is tuned to conduct both knowledge introspection (by imitating a few-shot GPT-3) and knowledge-grounded reasoning. We minimize two losses: a knowledge introspection loss

$$\mathcal{L}_{QK}(\theta) = -\log p_{QK}(k|q; \theta),$$

and a knowledge-grounded reasoning loss

$$\mathcal{L}_{QKA}(\theta) = -\log p_{QKA}(a^*|q, k; \theta),$$

where $k$ is a silver knowledge statement generated by the few-shot GPT-3. In training stage II, we follow the procedure in §2.3 to adapt the knowledge introspection and knowledge-grounded reasoning modes to each other.

## 2.6 Interleaved Optimization Schedule

Through empirical analysis, we have observed interleaving the two training losses in each stage

yields beneficial outcomes as opposed to jointly optimizing them. In stage I, we optimize $\mathcal{L}_{QK}$ for a specific number of iterations, followed by optimizing $\mathcal{L}_{QKA}$ for another set number of iterations, repeating this cycle. Similarly, In stage II, we optimize $\mathcal{L}_{PPO}$ for a designated number of iterations, followed by optimizing $\mathcal{L}_{QKA}$ for another set number of iterations, repeating this cycle. This design bears resemblance to the EM algorithm (Dempster et al., 1977), wherein the hidden variable corresponds to the knowledge statement. Optimizing $\mathcal{L}_{PPO}$ can be likened to estimating the hidden variables, while optimizing $\mathcal{L}_{QKA}$ is akin to updating the parameter estimation based on the current assignment of hidden variables. The interleaved optimization schedule is illustrated in Figure 2.

## 3 Experimental Setup

**Datasets.** To promote generalization, we train CRYSTAL on 10 datasets (following Liu et al. (2022a)): OpenBookQA (Mihaylov et al., 2018), ARC (easy and hard splits) (Clark et al., 2018), AI2Science (elementary and middle splits) (Clark et al., 2018), CommonsenseQA (Talmor et al., 2019), QASC (Khot et al., 2020), PhysicalIQA (Bisk et al., 2020), SocialIQA (Sap et al., 2019), and Winogrande (Sakaguchi et al., 2020). We use the development set of these datasets to evaluate model performance (i.e., *seen* evaluation). For *unseen* evaluation, we use the development set of 15 additional datasets: Com2Sense (Singh et al., 2021), SciQ (Welbl et al., 2017), QuaRel (Tafjord et al., 2019a), QuaRTz (Tafjord et al., 2019b), CycIC, ComVE (Wang et al., 2020), Winograd Schema Challenge (Levesque et al., 2011), COPA (Gordon et al., 2012), NumerSense (Lin et al., 2020), PROST (Aroca-Ouellette et al., 2021), SWAG (Zellers et al., 2018), HellaSwag (Zellers et al., 2019), CODAH (Chen et al., 2019), Story Cloze Test (Mostafazadeh et al., 2016), and $\alpha$NLI (Bhagavatula et al., 2020). See Table 9 (appendix) for details. On the training datasets, we get silver knowledge from the davinci version of GPT-3 (Brown et al., 2020), with the few-shot prompts in Liu et al. (2022a).

**Models.** Similar to Liu et al. (2022a), we initialize CRYSTAL with T5 (Raffel et al., 2020). The value regression head is initialized from scratch at the beginning of stage II training. We experiment with three model sizes: T5-large, T5-3b, and T5-11b. We train models on V100 GPUs (8

for T5-large, 16 for T5-3b, and 64 for T5-11b), with the Huggingface Transformers and Accelerate libraries (Wolf et al., 2019; Gugger et al., 2022). See Table 10 (appendix) for the complete hyperparameter settings.

**Baselines.** We primarily compare CRYSTAL with models trained on the same datasets using the standard QA objective (i.e., Equation 1), referred as "Direct QA". These models are also based on the pretrained T5 of the three sizes above. Additionally, we compare our model to Rainier (Liu et al., 2022a) and several CoT-distilled models. Among these, fine-tune-CoT (Ho et al., 2022) use zero-shot reasoning chains elicited from text-davinci-002 to finetune smaller variants of GPT-3; SCoTD (Li et al., 2023a) employs a similar distillation strategy, whereas the teacher model is code-davinci-002 and the target model is OPT up to 1.3B parameters; SCOTT (Wang et al., 2023) proposes additional techniques to improve the reasoning integrity, including a contrastive decoding method to elicit more consistent reasoning chains from the teacher model and a counterfactual reasoning method to train the target model, and yet does not enable full bidirectional adaptation between the reasoning process and the final prediction, as our method does. It is worth noting that these CoT-distilled models are often trained on specific datasets, so we only present their reported performance on the datasets they were trained on.

We also report the existing SOTA performance achieved by non-retrieval methods on each seen dataset.[2] We exclude retrieval-based methods for fair comparison, because CRYSTAL does not rely on retrieval from extra sources.

## 4 Results

### 4.1 Performance

The performance results are presented in Table 2 and Table 3. We organize the results based on the size of the models and compare them to baseline models that are no smaller than our models.

On seen datasets, across all model sizes we experiment with, CRYSTAL consistently outperforms the direct QA baseline (by 1.5%~2.5% depending on model size). This demonstrates that our training process is superior to the standard supervised training and brings substantial performance gains to the

---

[2]Accessed from the AI2 leaderboards on 08/24/2023.

| Method | Base model | Size | All | OBQA | ARC_e | ARC_h | AI2Sci_e | AI2Sci_m | CSQA | QASC | PIQA | SIQA | WG |
|---|---|---|---|---|---|---|---|---|---|---|---|---|---|
| SOTA (w/o retrieval) | – | – | – | 87.80 | – | 81.14 | – | – | 82.20 | 72.28 | 90.13 | 83.15 | 91.28 |
| Fine-tune-CoT | GPT-3-babbage | 1.3B | – | – | – | – | – | – | 43.08 | – | – | – | – |
| SCoTD | OPT-1.3b | 1.3B | – | 67.00 | – | – | – | – | 67.00 | – | – | – | – |
| Rainier (+ UnifiedQA) | T5-large | 770M | 62.58 | **69.60** | 67.72 | **55.18** | 68.29 | 63.20 | 67.24 | 54.97 | 65.67 | 57.01 | 56.91 |
| Direct QA | T5-large | 770M | 65.07 | 63.00 | 64.74 | 48.49 | **72.36** | 65.60 | 67.40 | 54.75 | 75.19 | 69.19 | 69.93 |
| CRYSTAL (ours) | T5-large | 770M | **66.74** | 64.20 | 65.61 | 52.84 | 71.54 | **68.00** | 70.52 | **56.80** | 75.68 | **69.81** | 72.38 |
| Fine-tune-CoT | GPT-3-curie | 6.7B | – | – | – | – | – | – | 56.76 | – | – | – | – |
| SCOTT | T5-3b | 3B | – | – | – | – | – | – | 75.40 | 65.00 | – | – | – |
| Direct QA | T5-3b | 3B | 75.84 | 72.00 | 77.19 | 63.55 | 83.74 | 75.20 | 76.99 | 67.82 | 83.03 | 76.77 | 82.08 |
| CRYSTAL (ours) | T5-3b | 3B | **78.33** | **74.20** | **78.25** | **66.22** | **84.55** | **79.20** | **80.10** | **74.30** | **84.49** | **78.40** | **83.58** |
| Direct QA | T5-11b | 11B | 82.49 | 80.00 | 84.56 | 72.91 | 87.80 | 84.00 | 81.98 | 78.29 | **88.36** | 78.45 | 88.56 |
| CRYSTAL (ours) | T5-11b | 11B | **84.58** | **85.40** | **87.54** | **73.24** | **89.43** | **84.80** | **82.31** | **81.97** | 88.08 | **82.24** | **90.77** |

Table 2: Results on seen datasets. Accuracy on the development set is reported.

| Method | Size | All | C2S | SciQ | QuaRel | QuaRTz | CycIC | ComVE | WSC | COPA | NumerSense | PROST | SWAG | HellaSwag | CODAH | SCT | aNLI |
|---|---|---|---|---|---|---|---|---|---|---|---|---|---|---|---|---|---|
| Direct QA | 770M | 60.93 | 55.75 | **66.80** | **71.22** | 67.45 | **49.83** | 83.45 | 78.02 | 74.20 | 23.00 | 38.07 | 46.48 | 45.65 | 62.36 | 86.37 | 65.27 |
| CRYSTAL (ours) | 770M | **62.95** | **56.52** | 66.70 | 70.86 | **67.71** | 49.72 | **85.46** | **81.68** | **77.60** | 23.00 | **41.34** | **51.54** | **48.51** | **66.43** | **90.27** | **66.84** |
| Direct QA | 3B | 67.73 | 58.57 | 75.90 | **81.29** | 70.83 | 55.35 | 93.08 | 82.05 | 90.20 | 20.00 | 36.83 | 57.77 | 49.31 | 75.54 | 95.14 | 74.15 |
| CRYSTAL (ours) | 3B | **72.06** | **65.98** | **79.50** | 80.58 | **73.96** | **60.20** | **95.49** | **89.74** | **91.20** | **28.50** | **44.35** | **60.48** | **56.58** | **81.45** | **96.37** | **76.57** |
| Direct QA | 11B | 76.83 | 75.45 | 83.20 | **86.69** | **77.34** | 69.57 | 96.89 | **94.14** | 94.00 | 30.50 | 44.39 | 65.68 | 69.37 | 84.73 | 97.70 | **82.83** |
| CRYSTAL (ours) | 11B | **80.37** | **85.93** | **85.30** | 85.97 | 76.30 | **70.89** | **98.09** | 93.77 | 94.00 | **41.50** | **59.37** | **69.58** | **76.06** | **87.64** | **98.50** | 82.64 |

Table 3: Results on unseen datasets. Accuracy on the development set is reported.

model. CRYSTAL also outperforms the combination of Rainier and UnifiedQA, especially over the last five datasets (which UnifiedQA is not trained on). This shows the benefit of adapting knowledge-grounded reasoning to the introspected knowledge.

CRYSTAL performs very closely to existing non-retrieval SOTA methods, setting new SOTA on two datasets (CommonsenseQA, QASC), and has less than 3% gap on other four (OpenBookQA, PIQA, SIQA, Winogrande). It is worth noting that these SOTA methods are good on different datasets, whereas CRYSTAL is a single model with strong performance on all these benchmarks. CRYSTAL is also competitive when compared with CoT-distilled models with similar sizes. The large and 3b versions of CRYSTAL beat Fine-tune-CoT on CommonsenseQA by 27% and 23%, respectively, despite having smaller model sizes. CRYSTAL-large is comparable to SCoTD on OpenBookQA and CommonsenseQA, and CRYSTAL-3b significantly outperforms SCOTT on CommonsenseQA (by 5%) and QASC (by 9%).

Being trained on multiple commonsense datasets, CRYSTAL exhibits good generalization to unseen datasets. As shown in Table 3, CRYSTAL achieves a 2.0%~4.3% average accuracy improvement over the direct QA baseline on the unseen evaluation benchmarks. The largest version of our

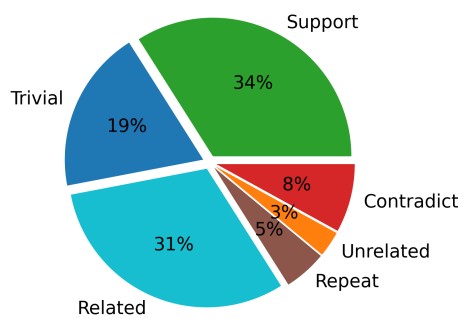

Figure 3: Expert annotation on the relationship between the introspected knowledge and the final prediction.

model, CRYSTAL-11b, achieves an average accuracy of over 80% on these benchmarks.

## 4.2 Interpretability

Beside QA accuracy, we measure whether the introspective reasoning conducted by CRYSTAL provides good interpretability to its reasoning process. We asked three NLP experts to annotate the relationship between the introspected knowledge and the final prediction. We randomly selected 100 examples (four from each of the 25 datasets, including both seen and unseen ones), and each annotator made a full pass over them. For each example, the annotator chooses one of the following labels:

| Task | Question / CRYSTAL's introspected knowledge | Direct QA's pred / CRYSTAL's pred |
|---|---|---|
| WG | They discussed the company's budget at the business meeting but the _ was boring and the topic of the budget ran long. (A) budget (B) meeting | A |
| | **The topic of the meeting was boring.** | **B** |
| PIQA | Find spices easily in the kitchen. (A) Arrange spices from hot to mild in the kitchen in order to find them by taste. (B) Arrange your spices alphabetically to make finding them easy. | A |
| | **A spice alphabet is used to find spices.** | **B** |
| QASC | What comes from cows? (A) pork (B) can be organic (C) holding nutrients (D) drinking water (E) rice (F) antigens (G) nutritious fluid (H) corn | A |
| | **Cows produce milk.** | **G** |
| CSQA | Paul wants carrots and doesn't need to drive anywhere. He gets them from where? (A) refrigerator (B) store (C) farmer's market (D) supermarket (E) dryer | D |
| | **Carrots are stored in the refrigerator.** | **A** |
| OBQA | Frilled sharks and angler fish live far beneath the surface of the ocean, which is why they are known as (A) Deep sea animals (B) fish (C) Long Sea Fish (D) Far Sea Animals | D |
| | **Deep sea animals are found in the ocean.** | **A** |
| ARC_e | An anemometer is a tool that measures (A) wind direction. (B) wind speed. (C) air pressure. (D) air temperature. | B |
| | **An anemometer measures wind speed and direction.** | **C** |

Table 4: Examples of CRYSTAL's introspected knowledge and predictions grounded in the knowledge. The first row of each section is the original question and the prediction made by the direct QA model; the second row is the knowledge statement generated by CRYSTAL in the knowledge introspection mode, and the prediction made by CRYSTAL under knowledge-grounded reasoning with this knowledge statement. We show correct answers in green and incorrect answers in red.

- **Support**: The knowledge can be part of a non-trivial reasoning chain that supports the predicted answer.

- **Trivial**: The knowledge is a trivial paraphrase of the question and the predicted answer.

- **Repeat**: The knowledge is a mere repetition of known information given in the question.

- **Related**: The knowledge is topically related to the question and/or the choices, but cannot be part of a reasoning chain to support or refute any of the choices.

- **Unrelated**: The knowledge is unrelated to the question.

- **Contradict**: The knowledge can be part of a reasoning chain that refutes the predicted answer, or supports a different choice.

See Table 11 (appendix) for a detailed description of these labels and some examples.

The annotators reached a moderate level of agreement (Fleiss $\kappa = 0.53$ (Landis and Koch, 1977)). As shown in Figure 3, in 34% of the cases the introspected knowledge is found to support the final prediction in a non-trivial manner. In 19% of

the cases the knowledge trivially entails the prediction, and in another 31% of the cases the knowledge is otherwise related to the question. The knowledge repeats known information in the question 5% of the time, and is unrelated to the question or contradicts with the prediction 11% of the time. Overall, the reasoning process has good interpretability in the majority of cases.

### 4.3 Qualitative Examples

We present several examples in Table 4 to illustrate the reasoning process of CRYSTAL. In most cases, the introspective reasoning carried out by CRYSTAL leads to more accurate predictions compared to the direct QA model. The knowledge introspected by CRYSTAL often proves to be beneficial in arriving at the correct prediction from human interpretation standpoint. For example, the knowledge *"Cow produce milk"* aids in concluding that *"Nutritious fluid comes from cows"* (with the implicit knowledge that *"Milk is nutritious fluid"*). This showcases how the knowledge-grounded reasoning of CRYSTAL leverages introspected knowledge to reach accurate predictions. However, there are exceptional cases where the knowledge-grounded reasoning fails to incorporate the introspected knowl-

| Method | # trained models | # frozen models | # forwards | # backwards | # optimizer steps |
|---|---|---|---|---|---|
| Rainier | 2 | 2 | $5 + 2s$ | $2s$ | $2s$ |
| CRYSTAL | 1 | 1 | $4 + s$ | $s$ | $s$ |

Table 5: Theoretical memory and time consumption of PPO training in CRYSTAL. $s$ is the number of minor steps in each PPO iteration. (In our experiments, we use $s = 4$.)

| Model | Base model | Trainable params | # GPUs | Total GPU mem | PPO training speed |
|---|---|---|---|---|---|
| Rainier | T5-large | 1.54B | 8 | 153 GiB | 10.97 s/it |
| CRYSTAL | T5-large | 770M | 8 | 129 GiB | 6.96 s/it |
| CRYSTAL | T5-3b | 3B | 16 | 488 GiB | 14.07 s/it |
| CRYSTAL | T5-11b | 11B | 64 | 2032 GiB | 60.30 s/it |

Table 6: Empirical memory usage and speed of training CRYSTAL (stage II). Experiments are conducted on V100 GPUs. Fully sharded data parallel (FSDP) and bfloat16 mixed precision are enabled.

edge. For example, the correct knowledge that *"An anemometer measures wind speed and direction"* is introspected, but CRYSTAL still predicts *"air pressure"* instead of *"wind speed"* as the thing measured by anemometers.

## 4.4 Memory and Time Efficiency

As mentioned in §2.4, the PPO training in CRYSTAL improves efficiency of the conventional PPO algorithm by model sharing. In this section, through theoretical and empirical analysis, we compare the memory and time consumption of PPO training in CRYSTAL and Rainier (Liu et al., 2022a), which employs the conventional PPO.

PPO in Rainier requires three different models: a policy model, a value model, and a reward model. The policy model is Rainier, while the reward model is a fixed QA model. The value model is a separate model that shares the same architecture as Rainier, with the exception that it has a value regression head instead of a language modeling head. The policy and value models are trained simultaneously, while the reward model remains frozen. Additionally, an initial version of the policy model must be retained (to calculate the KL penalty term). Therefore, a total of four models are stored, with two of them being actively updated. In each PPO iteration, Rainier requires $5 + 2s$ forward passes, $2s$ backward passes, and $2s$ optimizer updates on the model. ($s$ is the number of minor steps in each PPO iteration.) This involves executing a gradient-less rollout from the policy, one gradient-less forward pass on the value model, another gradient-less forward pass on the initial policy model, two gradient-less forward passes on the reward model, and for each minor step in the

iteration, conducting one forward-backward pass and one optimizer update on the policy model and the value model, respectively.

In contrast to Rainier, PPO on CRYSTAL needs to store only two models: a shared policy/value/reward model which is being actively updated, and an initial version of the policy model (to compute the KL penalty term). In each PPO iteration, CRYSTAL needs $4 + s$ forward passes, $s$ backward passes, and $s$ optimizer updates on the model: a gradient-less rollout from the policy, one gradient-less forward pass on the initial policy model, two gradient-less forward passes on the reward model, plus for each minor step in the iteration, one forward-backward pass and one optimizer update on the policy/value model.

Table 5 summarizes the theoretical memory and time consumption of Rainier and CRYSTAL, and Table 6 reports the empirical memory usage and speed in the stage II training of these models. Compared with Rainier, CRYSTAL has less trainable parameters, consumes less GPU memory, and has faster training speed. The superior memory and time efficiency of CRYSTAL enables training larger models, and a 11b model can be reinforced with 64 V100 GPUs.

## 4.5 Ablations

**The effect of RL.** We report the impact of removing RL (i.e. training stage II) in Table 7. Across different model sizes, the performance of CRYSTAL on seen datasets consistently decreases by approximately 0.5% to 0.6% when training stage II is omitted. This highlights the significance of RL in enhancing the knowledge introspection and knowledge grounded reasoning capability of CRYSTAL.

| Method | Size | Seen |
|---|---|---|
| CRYSTAL (ours) | 770M | **66.74** |
| - Stage II | 770M | 66.16 |
| + Direct QA loss | 770M | 65.36 |
| CRYSTAL (ours) | 3B | **78.33** |
| - Stage II | 3B | 77.79 |
| CRYSTAL (ours) | 11B | **84.58** |
| - Stage II | 11B | 84.08 |

Table 7: Ablations on the RL training stage (i.e., stage II). Average accuracy on the development set of seen datasets is reported.

| Stage I | Stage II | Seen |
|---|---|---|
| interleaved | interleaved | **66.74** |
| joint | interleaved | 66.66 |
| joint | joint | 66.31 |

Table 8: Ablations on the interleaved training objectives. Average accuracy on the development set of seen datasets is reported.

**Impact of the direct QA loss.** We experimented with training the stage I model with the addition of the direct QA loss (§2.3, Equation 1). As shown in Table 7, training with this additional loss hurts performance by 0.8%. We therefore did not include this loss in the training objective of CRYSTAL.

**Interleaved objectives.** To demonstrate the advantages of interleaving the training objectives, we explore an alternative approach using a joint objective. In this approach, during training stage I, we optimize the joint loss, $\mathcal{L}_{QK} + \mathcal{L}_{QKA}$, in each iteration. Similarly, during training stage II, we optimize the joint loss, $\mathcal{L}_{PPO} + \mathcal{L}_{QKA}$. Table 8 presents the results of this approach, where the interleaved objectives are replaced with the joint version. As such, the performance on seen datasets decreases. This suggests that the interleaving of objectives in CRYSTAL provides a benefit over the joint optimization approach.

## 5 Related Work

**Knowledge-augmented reasoning.** There has been numerous work that grounds reasoning in model-generated knowledge (Bosselut et al., 2021; Rajani et al., 2019; Latcinnik and Berant, 2020; Shwartz et al., 2020; Paranjape et al., 2021; Liu et al., 2022b; Gu et al., 2022; Liu et al., 2022a; Wang et al., 2022b,a; Yu et al., 2022; Li et al., 2023b; Wei et al., 2022). We summarize these

methods in Table 12 (appendix). Our method is the first to account for the mutual adaptation of knowledge generation and knowledge-grounded reasoning in a unified model setting.

**Relation to chain-of-thought distillation.** A series of work endow smaller LMs with step-by-step reasoning capabilities by distilling from chain-of-thought (CoT) generated by large LMs (Li et al., 2022; Shridhar et al., 2022; Magister et al., 2022; Ho et al., 2022; Fu et al., 2023; Li et al., 2023a; Wang et al., 2023). We share similarity with this line of work in that our part of training process (i.e., training stage I) include distilling the emergent capability of a larger LM to a smaller one. We differ in that we capture the introspective nature of knowledge required for commonsense reasoning, and we further use reinforcement learning to improve the synergy between reasoning paths and final answer predictions.

**Improving from self-feedback.** Several papers have proposed to improve LMs using feedback from themselves. For example, Zelikman et al. (2022) proposes to train a model with its self-generated reasoning steps that result in itself making the correct final predictions. Huang et al. (2022) chooses which self-generated reasoning chains to train on by selecting the high-confidence, self-consistent ones. Both papers use supervised loss to improve the model. To our best knowledge, we are the first to improve models from self-feedback using RL.

Concurrent to our work, Madaan et al. (2023) proposes an inference-time method to improve text generation by taking an LM's own feedback on the output, and yet it relies on the emergent behavior of LLMs, whereas CRYSTAL improves through RL and can be applied to smaller LMs to achieve higher performance than larger LLMs.

## 6 Conclusion

We develop a method to build introspective reasoners that achieves superior performance and good interpretability on commonsense reasoning tasks. Compared with prior literature, our method comprehensively accounts for the introspective nature of knowledge required in commonsense reasoning, and the mutual adaptation of knowledge introspection and knowledge-grounded reasoning. Our approach highlights the feasibility and benefit of training neural models with self-feedback.

## Limitations

CRYSTAL is intended to solve commonsense QA problems, and its performance on non-commonsense applications is unknown and thus requires further investigation. There is also a limit on the length of knowledge it generates in our experimental setting, and it has not been tested on generating long and coherent text. Extra care should be taken when applying our model in production environments, especially when making critical decisions or exposing its generated contents directly to human end users.

## Acknowledgements

We thank members of the H2lab for their constructive feedback. This work was funded in part by the DARPA MCS program through NIWC Pacific (N66001-19-2-4031), NSF IIS-2044660, NSF DMS-2134012, and ONR N00014-18-1-2826. JL is supported in part by the Meta AI Mentorship program.

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

## A More on Method

**I/O format of CRYSTAL.** The input/output format illustrated in Table 1 is adapted from Khashabi et al. (2020), based on which we made the following changes:

- The input is appended with a marker that indicates which mode of operation is desired. In knowledge introspection this marker is "Knowledge:", and in knowledge-grounded reasoning it is "Answer:".

- The output is the letter for the predicted answer choice, not the actual content of an answer choice.

- The input and output text are not lowercased.

## B More on Experimental Setup

Table 9 shows the datasets we use for training and evaluation, along with their citations. Table 10 reports the hyperparameters.

## C More on Results

Table 11 reports the detailed instructions for the human evaluation.

## D More on Related Work

Table 12 compares CRYSTAL with existing methods for knowledge-augmented commonsense reasoning.

| Abbr. | Name | Citation | Link |
|---|---|---|---|
| | | TRAINING + EVALUATION (SEEN) | |
| OBQA | OpenBookQA | Mihaylov et al. (2018) | https://github.com/allenai/unifiedqa |
| ARC_e | ARC (easy) | Clark et al. (2018) | https://github.com/allenai/unifiedqa |
| ARC_h | ARC (hard) | Clark et al. (2018) | https://github.com/allenai/unifiedqa |
| AI2Sci_e | AI2 Science (elem) | Clark et al. (2018) | https://github.com/allenai/unifiedqa |
| AI2Sci_m | AI2 Science (middle) | Clark et al. (2018) | https://github.com/allenai/unifiedqa |
| CSQA | CommonsenseQA | Talmor et al. (2019) | https://github.com/allenai/unifiedqa |
| QASC | QASC | Khot et al. (2020) | https://github.com/allenai/unifiedqa |
| PIQA | Physical IQA | Bisk et al. (2020) | https://github.com/allenai/unifiedqa |
| SIQA | Social IQA | Sap et al. (2019) | https://github.com/allenai/unifiedqa |
| WG | Winogrande | Sakaguchi et al. (2020) | https://github.com/allenai/unifiedqa |
| | | EVALUATION (UNSEEN) | |
| C2S | Com2Sense (paired) | Singh et al. (2021) | https://github.com/PlusLabNLP/Com2Sense/tree/master/data |
| SciQ | SciQ | Welbl et al. (2017) | https://allenai.org/data/sciq |
| QuaRel | QuaRel | Tafjord et al. (2019a) | https://allenai.org/data/quarel |
| QuaRTz | QuaRTz | Tafjord et al. (2019b) | https://allenai.org/data/quartz |
| CycIC | CycIC (mc) | – | https://leaderboard.allenai.org/cycic/submissions/get-started |
| ComVE | ComVE (task A) | Wang et al. (2020) | SemEval2020-Task4-Commonsense-Validation-and-Explanation |
| WSC | WSC | Levesque et al. (2011) | https://huggingface.co/datasets/winograd_wsc |
| COPA | COPA | Gordon et al. (2012) | https://huggingface.co/datasets/super_glue |
| NumerSense | NumerSense | Lin et al. (2020) | https://github.com/INK-USC/NumerSense/tree/main/data |
| PROST | PROST | Aroca-Ouellette et al. (2021) | https://huggingface.co/datasets/corypaik/prost |
| SWAG | SWAG | Zellers et al. (2018) | https://github.com/rowanz/swagaf/tree/master/data |
| HellaSwag | HellaSwag | Zellers et al. (2019) | https://github.com/rowanz/hellaswag/tree/master/data |
| CODAH | CODAH | Chen et al. (2019) | https://github.com/Websail-NU/CODAH/tree/master/data |
| SCT | Story Cloze Test | Mostafazadeh et al. (2016) | https://cs.rochester.edu/nlp/rocstories/ |
| $\alpha$NLI | $\alpha$NLI | Bhagavatula et al. (2020) | https://leaderboard.allenai.org/anli/submissions/get-started |

Table 9: Dataset details. We show the link from which we retrieved each dataset.

| Symbol | Value | Description |
|--------|-------|-------------|
| | SHARED HYPERPARAMETERS | |
| $L_Q$ | 256 | Max number of tokens in question (including choices). |
| $L_K$ | 32 | Max number of tokens in knowledge. |
| $L_A$ | 2 | Max number of tokens in answer. |
| | GETTING SILVER KNOWLEDGE FROM FEW-SHOT GPT-3 | |
| $M$ | 20 | Number of knowledge statements to sample from GPT-3, per question. |
| $p$ | 0.5 | Parameter for nucleus sampling from GPT-3. |
| | STAGE I: IMITATION LEARNING | |
| $B$ | 64 | Batch size for training. |
| $S$ | 50,000 | Total number of training iterations. |
| $S_{QK}$ | 500 | Number of iterations for knowledge introspection in each interleaving cycle. |
| $S_{QKA}$ | 500 | Number of iterations for knowledge-grounded reasoning in each interleaving cycle. |
| $\eta$ | $1 \times 10^{-5}$ | Learning rate of Adam optimizer. |
| | STAGE II: REINFORCEMENT LEARNING | |
| $\alpha$ | 1.0 | Weight of value model loss in PPO. |
| $\beta$ | 0.2 | Weight of entropy bonus term in reward. |
| $\gamma$ | 1.0 | Discount factor for rewards. |
| $\lambda$ | 0.95 | Parameter for advantage estimation. |
| $\varepsilon$ | 0.2 | Clipping range for the *clipped surrogate objective*. |
| $\tau$ | 0.7 | Temperature for knowledge sampling in PPO training. |
| $E$ | 2M | Total number of training episodes. |
| $B$ | 64 | Batch size for training. |
| $S$ | 31,250 | Total number of training iterations. |
| $s$ | 4 | Number of PPO update steps in each iteration for knowledge introspection. |
| $S_{PPO}$ | 500 | Number of iterations for knowledge introspection in each interleaving cycle. |
| $S_{QKA}$ | 500 | Number of iterations for knowledge-grounded reasoning in each interleaving cycle. |
| $\eta$ | $1 \times 10^{-5}$ | Learning rate of Adam optimizer (for CRYSTAL-large). |
| | $1 \times 10^{-6}$ | (for CRYSTAL-3b and -11b). |
| | INFERENCE | |
| $M$ | 10 | Number of knowledge statements to sample from CRYSTAL, per question. |
| $p$ | 0.5 | Parameter for nucleus sampling from CRYSTAL. |

Table 10: Hyperparameter settings.

| Label | Description | Example |
|---|---|---|
| Support | The knowledge can be part of a non-trivial reasoning chain that supports the predicted answer. | **Question:** Who watches a play in an auditorium? \n (A) building (B) crowd (C) city (D) group (E) high school
**Knowledge:** Audiences watch plays in auditoriums.
**Prediction:** (B) |
| Trivial | The knowledge is a trivial paraphrase of the question and the predicted answer. | **Question:** An alpha particle, which is emitted during alpha decay, consists of two protons and what else? \n (A) two neutrons (B) two nuclei (C) two positrons (D) two electrons
**Knowledge:** Alpha particles are composed of two protons and two neutrons.
**Prediction:** (A) |
| Repeat | The knowledge is a mere repetition of known information given in the question. | **Question:** The movement of crustal plates results from circulating currents in material beneath the crust of Earth. Which best describes the material which moves the crustal plates? \n (A) hot water (B) molten rock (C) liquid metal (D) solid iron
**Knowledge:** The movement of crustal plates is caused by circulating currents in material beneath the crust of Earth.
**Prediction:** (B) |
| Related | The knowledge is topically related to the question and/or the choices, but cannot be part of a reasoning chain to support or refute any of the choices. | **Question:** How are the particles in a block of iron affected when the block is melted? \n (A) The particles gain mass. (B) The particles contain less energy. (C) The particles move more rapidly. (D) The particles increase in volume.
**Knowledge:** Iron particles are affected by heat.
**Prediction:** (C) |
| Unrelated | The knowledge is unrelated to the question. | – |
| Contradict | The knowledge can be part of a reasoning chain that refutes the predicted answer, or supports a different choice. | **Question:** I need what to calculate the length from my big toe to my little toe? \n (A) Calculator (B) Tape Measure (C) A Graph (D) A Microscope
**Knowledge:** A calculator is used to calculate lengths.
**Prediction:** (B) |

Table 11: Instruction for the human evaluation.

| Method | Citation | Unified model | KG => KR | KR => KG |
|---|---|:---:|:---:|:---:|
| DynaGen | Bosselut et al. (2021) | ✓ | ✗ | ✗ |
| CAGE | Rajani et al. (2019) | ✗ | ✓ | ✗ |
| ST-GS | Latcinnik and Berant (2020) | ✗ | ✓ | ✓ |
| Self-talk | Shwartz et al. (2020) | ✓/ ✗ | ✗ | ✗ |
| Contrastive Expl. | Paranjape et al. (2021) | ✗ | ✓ | ✗ |
| GKP | Liu et al. (2022b) | ✓/ ✗ | ✗ | ✗ |
| DREAM | Gu et al. (2022) | ✗ | ✗ | ✗ |
| Rainier | Liu et al. (2022a) | ✗ | ✗ | ✓ |
| ALEAP | Wang et al. (2022b) | ✗ | ✓ | ✓ |
| PINTO | Wang et al. (2022a) | ✗ | ✓ | ✗ |
| GenRead | Yu et al. (2022) | ✓ | ✗ | ✗ |
| DSP | Li et al. (2023b) | ✗ | ✗ | ✓ |
| CoT | Wei et al. (2022) | ✓ | ✗ | ✗ |
| CRYSTAL | ours | ✓ | ✓ | ✓ |

Table 12: Comparison of existing knowledge-augmented reasoning methods. **Unified model**: if the method employs a unified model (rather than separate) for knowledge generation and knowledge-grounded reasoning. **KG => KR**: if the knowledge-grounded reasoning is trained to adapt to knowledge generation. **KR => KG**: if the knowledge generation is trained to adapt to knowledge-grounded reasoning.