# OpenReview forum: "Crystal: Introspective Reasoners Reinforced with Self-Feedback"
_EMNLP/2023/Conference — EMNLP 2023 Main_

### Official Review · Reviewer_s1EV · 2023-08-01

**Soundness:** 3

**Excitement:**

3: Ambivalent: It has merits (e.g., it reports state-of-the-art results, the idea is nice), but there are key weaknesses (e.g., it describes incremental work), and it can significantly benefit from another round of revision. However, I won't object to accepting it if my co-reviewers champion it.

**Paper Topic And Main Contributions:**

The paper aims to solve commonsense reasoning tasks, where additional knowledge is required. They adopt a unified model that operates on knowledge introspection and knowledge-grounded reasoning modes. To ensure the generated knowledge is effective, they adopt RL to encourage the model to generate knowledge statements that help the model predict the correct answers. To ensure the knowledge is leveraged for reasoning, they train the model to maximize the probabilities of the correct answers when knowledge is provided. Two training stages are conducted iteratively. Experiments demonstrate the effectiveness of the proposed model in achieving better results than the baselines in both seen and unseen tasks. The human evaluation shows the informativeness of the generated knowledge.

**Questions For The Authors:**

1.	How does your method improve the usefulness of the knowledge compared to Rainier? Can you also conduct the human evaluation on the generated knowledge from Rainier?
2.	How does training the model with standard supervision guarantee the model would leverage the knowledge for reasoning?


**Reasons To Accept:**

1.	The mutual adaptation of knowledge generation and grounding is an important research problem as it helps reveal the reasoning process of an otherwise blackbox model.
2.	The proposed training objectives and iterative training process are both novel and intuitive.
3.	The work carefully examines the method on a wide range of tasks, demonstrating the generalization of the method.


**Reasons To Reject:**

1.	It is unclear how the standard supervision would ensure the model indeed learns to leverage the knowledge instead of ignoring it.
2.	It seems the RL part of the work is largely based on the cited work Rainier, which limits the novelty of the work.


**Reproducibility:**

4: Could mostly reproduce the results, but there may be some variation because of sample variance or minor variations in their interpretation of the protocol or method.

**Reviewer Confidence:**

4: Quite sure. I tried to check the important points carefully. It's unlikely, though conceivable, that I missed something that should affect my ratings.

---

> ### Author Rebuttal · Authors · 2023-08-25
>
> Thanks for the insightful feedback! We really appreciate your recognition of the importance of the research problem we study.
>
> Below is our response to your concerns:
> 1. **Leveraging knowledge:** We encourage the model to leverage the introspected knowledge through our training objective design:
>     * More specifically, in training stage I, the model is trained on two losses: a knowledge introspection loss $L_\text{QK} = -\log p(k | q)$ and a knowledge-grounded reasoning loss $L_\text{QKA} = -\log p(a^* | q, k)$ (Sec 2.5). Hypothetically we could introduce a third loss: a straightforward QA loss, $L_\text{QA} = -\log p(a^* | q)$, but this would allow the model to take shortcuts and ignore the knowledge. We ensure knowledge is leveraged by **not** including this QA loss. We apply the same knowledge-grounded reasoning loss $L_\text{QKA}$ in stage II training.
>     * In early experiments, we found that including this QA loss hurts performance, probably by allowing the model to take shortcuts around the knowledge. We conducted this experiment with the Stage I trained model based on T5-large, so the baseline we compare with is Appendix Table 10 Line 2. Including this QA loss reduces accuracy from 66.16 down to 65.36, which motivated us to not include the QA loss. We will report this result in the updated paper.
>     * If we only use the QA loss, this is equivalent to our “Direct QA” baseline, and as reported in Table 2 & 3, this baseline underperforms our Crystal model. So to achieve better QA performance, it is in the best interest of the model to leverage the introspected knowledge.
>     * We do not enforce the knowledge to be leveraged when the model makes final predictions. If the model believes that the knowledge is wrong and implies a wrong answer, it is welcome to contradict the knowledge. In fact, as shown in our human evaluation (Sec 4.2), there is a small percentage of cases (8%) where the predicted answer contradicts the knowledge. However, in the majority of cases (53%) we found that the knowledge is directly indicative of the prediction.
> 2. **Novelty:** While we inherit the RL formulation and algorithm from the Rainier paper, our paper carries substantial novelty. (1) We design Crystal as a self-contained introspective reasoning model that learns from its self-feedback using reinforcement learning, which is novel compared to Rainier and other existing literature. (2) We show that it is viable and more efficient to use a single model as policy and feedback provider in PPO. (3) We also propose novel training objectives and optimization schedules to effectively prevent catastrophic forgetting due to this model sharing.
> 3. **Comparing usefulness of knowledge:** We did not compare our human evaluation with that reported in Rainier due to the difference in evaluation scheme. We asked if the knowledge is indicative of **the model’s final prediction**, whereas in Rainier the authors asked if the knowledge is indicative of **the correct answer**. We choose our scheme and because this research question is more helpful for understanding the interpretability of reasoning offered by our model.

---

### Official Review · Reviewer_Mz6D · 2023-08-05

**Soundness:** 4

**Excitement:**

3: Ambivalent: It has merits (e.g., it reports state-of-the-art results, the idea is nice), but there are key weaknesses (e.g., it describes incremental work), and it can significantly benefit from another round of revision. However, I won't object to accepting it if my co-reviewers champion it.

**Missing References:**

Some of these works maybe concurrent to this work, but it would be good to contrast crystal with Self-Refine (concurrent) and RL4F like paradigms
1. Afra Feyza Akyurek, Ekin Akyurek, Ashwin Kalyan, Peter Clark, Derry Tanti Wijaya, and Niket Tandon. 2023. RL4F: Generating Natural Language Feedback with Reinforcement Learning for Repairing Model Outputs. In Proceedings of the 61st Annual Meeting of the Association for Computational Linguistics (Volume 1: Long Papers), pages 7716–7733, Toronto, Canada. Association for Computational Linguistics.
2. Self-Refine: Iterative Refinement with Self-Feedback. Madaan et. al, 2023

**Paper Topic And Main Contributions:**

This paper proposes Crystal, a method that first introspects knowledge, and then generates an answer that is conditioned on the introspected knowledge. This introspected knowledge is then treated as "explanations" for the given prediction. Feedback on generating good quality explanations conditions on the answer and vice version is setup using an RL-like paradigm (similar to the Rainier work).

**Questions For The Authors:**

1. It is unclear whether there are any baselines where the knowledge introspection is trained on gold-set rationales (some datasets do have them, OpenBookQA for example). From what I could notice, silver set rationales from GPT-3 Does the quality of these reference rationales during training of the initial introspection model affect the eventual ceiling of performance that is reachable by Crystal?


**Reasons To Accept:**

The paper is extremely thorough with its experiments, including the diversity in datasets. The motivation for interdependence on the knowledge introspection with the answer generation is very relevant to current research directions on faithfulness of generated explanations with their answers. The human evaluations also further reinforce the association between the generated introspections with the label.

**Reasons To Reject:**

RR1 Comparisons with some more fundamental baselines - While the paper compares Crystal to CoT distilled approaches, another line of close work in the introspect then predict paradigm are retrieval augmented generators. Furthermore, it would be good to compare Crystal with standard CoT approaches.

**Reproducibility:**

4: Could mostly reproduce the results, but there may be some variation because of sample variance or minor variations in their interpretation of the protocol or method.

**Reviewer Confidence:**

4: Quite sure. I tried to check the important points carefully. It's unlikely, though conceivable, that I missed something that should affect my ratings.

---

> ### Author Rebuttal · Authors · 2023-08-25
>
> Thanks for the insightful feedback! We really appreciate your recognition of the thoroughness in our experiments and the value in our human evaluation.
>
> Below is our response to your concerns:
> 1. **Retrieval-augmented baselines:** In this paper we did not explicitly compare with knowledge-retrieval baselines because our method is most closely related to fully self-contained neural reasoners (i.e., no external sources at inference time). Prior work (Liu et al., 2022; Yu et al., 2022) have found that the performance of knowledge-retrieval methods can vary greatly with the quality of knowledge base, and model-generated knowledge is a more robust and scalable way to do knowledge-augmented reasoning.
> 2. **CoT baselines:** Thanks for the suggestion, we will add the standard CoT baseline in our next paper update. For your reference, we have evaluated the largest GPT-3.5 under the zero-shot setting (no CoT), and it underperforms our Crystal model:
> | Model | Size | Avg acc on seen benchmarks |
> | --- | --- | --- |
> | Crystal-11b | 11B | 84.58 |
> | text-davinci-003 | 175B | 79.03 |
>
>     * (With CoT we expect GPT-3.5 would perform better than this baseline.)
> 3. **Training on gold rationales:** In early experiments we tried replacing GPT3-generated silver knowledge with gold annotated rationales, and it did not lead to improved performance. Specifically, when replicating the supervised model in Rainier, we use the annotated rationales in ECQA (for CommonsenseQA) and QASC, and it did not improve the usefulness of model-generated knowledge over the original supervised model. In addition, gold annotated rationales do not exist for many other datasets we are interested in (e.g., Physical IQA, Social IQA), while GPT3-generated silver knowledge offers more flexibility so we can apply the knowledge introspection paradigm to more tasks.
> 4. **Missing references:** Thanks for sharing the related concurrent work, we provide contrast with these paradigms as follows (and will add to our next paper update):
>     * **Self-Refine:** It shares similarities with Crystal in that both improve text generation from the model’s own feedback. The difference is that Self-Refine is a purely inference-time method and relies on the emergent behavior of LLMs, whereas Crystal improves through reinforcement learning and can be applied to smaller LMs to achieve higher performance than larger LLMs.
>     * **RL4F:** It uses reinforcement learning to optimize a critique model so that the generated critique can most effectively correct the outputs of a frozen task model. In some sense RL4F is more similar to Rainier, if we draw an analogy between RL4F’s “critique” and Rainier’s “knowledge”, and between RL4F’s “task model” and Rainier’s “QA model”. Contrast to RL4F and Rainier, in Crystal we use a single model for the task and critique, and these two aspects are both trained to co-adapt to each other.
>
> References
> * Jiacheng Liu, Alisa Liu, Ximing Lu, Sean Welleck, Peter West, Ronan Le Bras, Yejin Choi, and Hannaneh Hajishirzi. 2022b. Generated knowledge prompting for commonsense reasoning. In Proceedings of the 60th Annual Meeting of the Association for Computational Linguistics (Volume 1: Long Papers), pages 3154–3169, Dublin, Ireland. Association for Computational Linguistics.
> * W. Yu, Dan Iter, Shuohang Wang, Yichong Xu, Mingxuan Ju, Soumya Sanyal, Chenguang Zhu, Michael Zeng, and Meng Jiang. 2022. Generate rather than retrieve: Large language models are strong context generators. ArXiv preprint, abs/2209.10063.

---

### Official Review · Reviewer_PuFC · 2023-08-05

**Soundness:** 4

**Excitement:**

4: Strong: This paper deepens the understanding of some phenomenon or lowers the barriers to an existing research direction.

**Paper Topic And Main Contributions:**

The research paper addresses the challenging problem of commonsense reasoning by proposing a novel solution called CRYSTAL, an introspective commonsense reasoner. CRYSTAL employs a two-step approach, where it first introspects knowledge statements relevant to a given question and then utilizes this knowledge to provide an answer. The introspection and reasoning processes in CRYSTAL are refined using reinforcement learning techniques.

The evaluation of CRYSTAL's performance consists of two main components. Firstly, the authors present the main results in Tables 2 and 3, which demonstrate that CRYSTAL outperforms both the standard supervised fine-tuning method (referred to as DirectQA) and chain-of-thought distilled methods across a diverse set of commonsense QA tasks. Secondly, the authors assess the quality of the introspected knowledge by obtaining expert annotations. They show that the knowledge generated by CRYSTAL proves to be valuable for the reasoning process, leading to correct answers in a significant proportion of cases. Specifically, in 34% of instances, the generated knowledge directly supports the answer, further validating the effectiveness of CRYSTAL's knowledge introspection process.

The paper contributes to the existing body of research on commonsense reasoning, presenting clear and well-supported claims. The technical details are extensively presented, enabling a comprehensive understanding of the proposed method.

However, one major concern with the paper lies in the evaluation section, specifically related to the DirectQA baseline model. Despite multiple readings of Section 3, the specific model used as DirectQA is not explicitly specified. It is crucial to have a meaningful baseline, ideally a state-of-the-art model from the leaderboard, to enable readers to draw informed conclusions regarding CRYSTAL's strengths and weaknesses in comparison. While it is not expected for the CRYSTAL to surpass the state-of-the-art performance on all tasks, reporting a well-defined baseline would aid in better understanding the advancements achieved by CRYSTAL in the context of the broader field of commonsense reasoning. This issue significantly affects my score for this paper, however, I am open to improving my score upon further clarification from the authors during rebuttal.

**Reasons To Accept:**

- Interesting problem
- Solid results

**Reasons To Reject:**

- Missing details in the evaluation
- Unclear baseline comparison

**Reproducibility:**

4: Could mostly reproduce the results, but there may be some variation because of sample variance or minor variations in their interpretation of the protocol or method.

**Reviewer Confidence:**

3: Pretty sure, but there's a chance I missed something. Although I have a good feel for this area in general, I did not carefully check the paper's details, e.g., the math, experimental design, or novelty.

---

> ### Author Rebuttal · Authors · 2023-08-25
>
> Thanks for the insightful feedback! We really appreciate your recognition of the clarity and detailedness of our presentation.
>
> Below is our response to your concerns:
> 1. **The “Direct QA” baseline:** As we describe in Sec 3 (Line 368), the “Direct QA” baseline is trained on the same datasets as Crystal, using a standard QA objective. We would like to additionally clarify that the base model is also the pretrained T5 models, of sizes ranging from T5-large to T5-11b, which is identical to the base model for Crystal. We compare the “Direct QA” baseline with our Crystal model with the same sizes (e.g., large vs large, 11b vs 11b), as shown in Table 2. We apologize for the missing details and we will add this to our next paper update.
>     * The QA objective used in “Direct QA” is formally: $\mathcal{L}_\text{QA} = -\log p(a^* | q)$
> 2. **SOTA performance**: Thanks for the suggestion, we surveyed the SOTA methods (w/o retrieval, for fair comparison with Crystal) and as below we report on seven datasets whose leaderboards are publicly available. On these datasets, our model Crystal is very close to the average SOTA accuracy (<1% gap, with new SOTA on 2 datasets). Note that these SOTA methods are good on different datasets, whereas Crystal is a single model with strong performance on all these benchmarks.
> | Model | Avg | OBQA | ARC_h | CSQA | QASC | PIQA | SIQA | WG |
> | --- | --- | --- | --- | --- | --- | --- | --- | --- |
> | Direct QA | 81.22 | 80.00 | 72.91 | 81.98 | 78.29 | 88.36 | 78.45 | 88.56 |
> | Crystal | 83.43 | 85.40 | 73.24 | **82.31** | **81.97** | 88.08 | 82.24 | 90.77 |
> | SOTA (w/o retrieval) | **84.00** | **87.80** | **81.14** | 82.20 | 72.28 | **90.13** | **83.15** | **91.28** |
> 3. **Evaluation details:** Aside from the insufficient description of the “Direct QA” baseline, we have provided comprehensive details about our experiments and evaluation in our submission. Please find additional details in the Appendix, where Appendix B documents the datasets we use and model hyperparameters, and Appendix C offers method ablations as well as more details on human evaluation. If there are any additional details that are missing, please definitely let us know and we are delighted to add those to the paper.

---

### Meta-Review · Area_Chair_CMcz · 2023-09-19

**Recommendation:** 3

**Metareview:**

The paper proposes a method Crystal that introspects knowledge statements related to a given commonsense question and uses this knowledge to provide an answer.  Crystal considers two steps: (a) introspective reasoning and (b) knowledge introspection. The idea of knowledge introspection is similar to Rainier's framework.  Crystal extends this idea to a unified introspective reasoning model.


Pros:
- The paper is well-written and thorough with every detail.
- The idea of optimising the knowledge generator and reasoner in an interleaved fashion is interesting.
- The qualitative and quantitative analysis is solid.


Cons:
- The main concern is that the baseline is weak. Comparison with only DirectQA does not give the full picture since it doesn’t use any knowledge. An ideal comparison should be:
 L_QA = -logp(a*|q,k) where k is the silver knowledge generated by GPT-3 (Table 3). This will show how the overall model improved knowledge quality and reasoning capability.  In Lines 288-289, it is unclear which version of the GPT-3 model is used to generate silver knowledge. Is it the same knowledge as Rainier's paper?  It will make the results in Table 2 more reliable.

Finally, The AreaChair appreciate the authors' response to clarify the doubts of the reviewers.

---

### Decision · Program_Chairs · 2023-10-07

**Decision:**

Accept-Main

**Comment:**

The paper proposes a method Crystal that introspects knowledge statements related to a given commonsense question and uses this knowledge to provide an answer.  Crystal considers two steps: (a) introspective reasoning and (b) knowledge introspection. The idea of knowledge introspection is similar to Rainier's framework.  Crystal extends this idea to a unified introspective reasoning model.


Pros:
- The paper is well-written and thorough with every detail.
- The idea of optimising the knowledge generator and reasoner in an interleaved fashion is interesting.
- The qualitative and quantitative analysis is solid.


Cons:
- The main concern is that the baseline is weak. Comparison with only DirectQA does not give the full picture since it doesn’t use any knowledge. An ideal comparison should be:
 L_QA = -logp(a*|q,k) where k is the silver knowledge generated by GPT-3 (Table 3). This will show how the overall model improved knowledge quality and reasoning capability.  In Lines 288-289, it is unclear which version of the GPT-3 model is used to generate silver knowledge. Is it the same knowledge as Rainier's paper?  It will make the results in Table 2 more reliable.

Finally, The AreaChair appreciate the authors' response to clarify the doubts of the reviewers.